behaviour/cognition/developmental biology

rhythm, spontaneous motor tempo, infants, locomotion, development

**Author for correspondence:**
Sinead Rocha
e-mail: sineadrocha@gmail.com

# Rate of infant carrying impacts infant spontaneous motor tempo

Sinead Rocha[1,2], Victoria Southgate[3] and Denis Mareschal[2]

[1]Department of Psychology, University of Cambridge, Cambridge CB2 1TN, UK
[2]Birkbeck University of London, London WC1E 7HX, UK
[3]Department of Psychology, University of Copenhagen, Kobenhavn, Denmark

SR, 0000-0001-5231-9062; DM, 0000-0002-9828-9548

Rhythm production is a critical component of human interaction, not least forming the basis of our musicality. Infants demonstrate a spontaneous motor tempo (SMT), or natural rate of rhythmic movement. Here, we ask whether infant SMT is influenced by the rate of locomotion infants experience when being carried. Ten-month-old, non-walking infants were tested using a free drumming procedure before and after 10 min of being carried by an experimenter walking at a slower (98 BPM) or faster (138 BPM) than average tempo. We find that infant SMT is differentially impacted by carrying experience dependent on the tempo at which they were carried: infants in the slow-walked group exhibited a slower SMT from pre-test to post-test, while infants in the fast-walked group showed a faster SMT from pre-test to post-test. Heart rate data suggest that this effect is not due to a general change in the state of arousal. We argue that being carried during caregiver locomotion is a predominant experience for infants throughout the first years of life, and as a source of regular, vestibular, information, may at least partially form the basis of their sense of rhythm.

## 1. Introduction

One of the most captivating aspects of music is that it makes us want to move [1]. Intriguingly, movement also biases our perception of rhythm, with this bidirectional relationship between auditory rhythms and movement well documented from early in life [2–4]. Investigation of the ontogeny of rhythmic movement may offer unique insights into the origins of this bidirectional relationship. While the ability to synchronize one's movement to music develops slowly across childhood (see [5]), infants from five months of age can produce

a spontaneous motor tempo (SMT, or natural rate of rhythmic movement) [6]. The current paper asks if the link between auditory and gross motor rhythm can be in part explained by the prevalent behaviour of infant carrying.

Infant carrying may hold vital clues to the developmental origin of our rhythmic preferences. It has been widely noted that the tempo of most music and human locomotion coincide [7–12]. Trevarthen [13] argues that human bipedal locomotion provides our species with a rhythm, with regular footfalls forming an underlying beat to daily life. Prenatally, infants can hear the mother's footfall as well as experience the corresponding movement from within the womb. Parncutt [14] suggests that prenatal conditioning of the contingency of these two signals via locomotion is responsible for the link between auditory rhythm and body movement. Teie [15] proposes a direct mapping of prenatal experience of the mother's walking to the musical property of 'pulse', or beat. Critically, the *multimodal* nature of locomotive experience (as compared, for example, to fetal auditory perception of maternal heart rate or respiration) distinguishes locomotion from other rhythmic prenatal experiences [16].

Support for the idea that the experience of rhythmic walking impacts SMT comes from studies correlating body size to rhythmic preferences in adults. The rate at which an individual walks is in part determined by the length of their limbs and other such physical features, due to the pendulum nature of the leg swing, and aerobic cost optimization that depends on stature (e.g. [17]). Anthropometrics correlate with one's SMT, as measured by finger tapping [7], and full-body dancing [18], as well as preferred tempo in a perceptual listening task [19]. The regular vertical 'bounce' of stepping may be especially important for our rhythmic preferences: in studies of natural movement to music, there is evidence that different body parts group into different eigenmovements, such that the beat level of music tends to be represented over the body in vertical actions, especially of the torso [20,21], and it has long been proposed that the core of the sensory–motor relationship ubiquitous in rhythm may be primarily vestibular in nature [22,23].

For infants, who cannot walk independently for a protracted period of postnatal development, the experience of being carried by the locomoting caregiver likely provides critical further conditioning of paired auditory and motor rhythmic stimulation. In contemporary hunter–gatherer societies, infants are held or carried for most of the day [24]. It is well established that the newborn responds to vestibular–tactile–somatosensory rhythms (for a review, see [25]), and vestibular information impacts rhythm perception from infancy [2–4]. Such stimulation may have large-scale and long-lasting effects: Ayres [8] analysed 54 traditional, historically independent societies, and found that societies where infants are carried, either in a sling or on the body without support, produce a higher percentage of music with a regular rhythm than societies that use a cradle, cradleboard or hammock. Ayres [8] concludes that societal preference for a regular rhythm is correlated with the nature and frequency of infant carrying in that population.

In a cross-sectional study of infant drumming over the first years of life, Rocha *et al.* [6] found that infant SMT was predicted by parental body size, but not infant's own body size. Specifically, parent height predicted infant SMT such that infants with taller parents showed a slower SMT than infants with shorter parents. Infants provide a crucial test case for the experience of locomotion being a core driver of rhythmic preference. While a correlation between own body size and natural rate of movement in adulthood could be explained as the product of biomechanical resonance, or a natural frequency of movement across the body, the existence of a relationship between parent body size and infant SMT supports the idea that *experience* of locomotion—in this case when carried by the caregiver—drives our basic rhythmic preferences. Accordingly, while the dependent variables in Rocha *et al.* [6] were anthropometric indices, results were interpreted as indicative of the vast amounts of information gained by infants when they are carried by their caregiver, at the caregiver's walking cadence.

To date, it has not been directly tested whether the experience of being carried during locomotion impacts infant rhythm production, though seminal work shows even 2 min of vestibular movement can bias rhythm perception [2]. The current study aims to test the impact of caregiver locomotion on infant SMT, by experimentally manipulating the timing of locomotion that infants experience in a pre-test, training, post-test design. Infant SMT is measured at pre- and post-test using the free drumming task employed in Rocha *et al.* [6]. During the training period, non-walking, ten-month-old infants are carried in a forward-facing baby carrier and walked by the experimenter at either a Fast (138 BPM) or Slow (98 BPM) pace, for 10 min. Our primary hypothesis is that infants in the Fast condition will drum faster at post-test than pre-test, while infants in the Slow condition will drum more slowly at post-test than pre-test. In order to ensure that changes in SMT in the current study are not the

product of a general change in arousal, we additionally measure infant heart rate while at rest, immediately before and after carrying, and predict no change from pre- to post-test.

Additional measures of parent and infant anthropometrics were taken, as well as parent SMT as measured via free drumming, free tapping and a free treadmill walking measure. In line with Rocha *et al*. [6], we hypothesize that parental body size will predict infant SMT, such that infants with a larger parent will drum more slowly at pre-test than infants with a smaller parent. We further hypothesize that parent body size will predict parent's own SMT and walking cadence. Finally, we used the questionnaire from Rocha *et al*. [6] as a measure of infant gross motor abilities for exploratory analyses. While no differences were found in tempo or regularity of drumming by motor ability in the prior cross-sectional study, here we ask whether differences in motor experience in the current tightly controlled single-age sample would influence SMT.

# 2. Method

## 2.1. Participants

Forty-seven ten-month-olds took part in the study, in a between-subjects design (22 female; mean age = 10 months ($M$ = 305 days, range = 290 to 332 days)). Twenty-four infants ($M$ = 304 days, range = 291 to 323 days) were randomly allocated to the Fast condition, and 23 infants ($M$ = 305 days, range = 290 to 332 days) to the Slow condition. Infants had to provide at least one bout of drumming at both pre- and post-test for inclusion in the intervention analyses. Sixteen infants did not do so, which is in line with the attrition rate of infant studies [26]. A further two infants were excluded for sporadic, arhythmic, drumming, using the exclusion criteria set in Rocha *et al*. [6], further detail below. The intervention analysis therefore was composed of 15 infants in the Fast condition ($M$ = 304 days, range = 291 to 322 days) and 14 infants in the Slow condition ($M$ = 305 days, range = 291 to 332 days). In the analyses of anthropometric/questionnaire data, further infants/caregivers who provided valid data for each measure were included; corresponding degrees of freedom are provided. Only non-walking infants were recruited for the study. Five of the participating caregivers were male. All caregivers gave written, informed consent concerning the experimental procedure for themselves and their infant. Infants received a certificate and a t-shirt as a thank you for participation.

## 2.2. Procedure

We employed a pre-test, training, post-test design. Infants participated in a free drumming measure of SMT pre- and post-experience of the Fast (138 BPM) or Slow (98 BPM) walking conditions. The Fast and Slow rates reflect the extreme values of the normal range of cadence in free-speed walking for females aged 18–49 years [27]. Caregivers were subsequently asked to complete additional parental measures.

### 2.2.1. Pre- and post-test measure of spontaneous motor tempo

Infants were seated on a cushion adjacent to the caregiver or on the caregiver's lap. A 12-inch drum supported on an adjustable height table was placed over the infant's lap. To familiarize the infant with the instrument, the experimenter demonstrated that the drum produces noise, telling the infant 'Look!' and then hitting the drum once. If the infant did not spontaneously try to drum herself, the experimenter repeated the demonstration, leaving at least two seconds between each demonstration. In this way, infants were not primed with a rate at which to hit the drum.

The trial started when the experimenter commenced the demonstration and lasted for 5 min. Infants were congratulated when they hit or interacted with the drum. If infants moved away from the drum, the caregiver returned the infant to their seated position. At the end of the trial, infants were congratulated again.

### 2.2.2. Carrying experience—Fast condition

The caregiver placed the infant in a forward-facing infant sling worn by the experimenter. The sling supported the infant's weight so that the experimenter had both hands free. The experimenter, with the infant, stepped onto the treadmill. A display monitor facing the infant from a distance of 50 cm was turned on and displayed an infant cartoon. The experimenter remained stationary for 1 min, and infant and experimenter heart rate was recorded using surface electromyography (EMG). Following

the heart rate recording, the experimenter started the treadmill and gradually increased the speed for up to 1 min until it reached a comfortable speed at which to walk at 138 BPM (434 ms between steps). In order to keep pace while walking the experimenter listened to a metronome recording at 138 BPM through one in-ear headphone. The experimenter walked with both hands holding the handlebars and with easy access to the speed controls and emergency stop. The experimenter walked on the treadmill for 10 min. During the training, the caregiver was seated adjacent to the treadmill. Both experimenter and caregiver spoke to the infant in the first instance if the infant was not engaged with the video presented to them. An assistant also blew bubbles and provided toys if the infant became unsettled. At the end of the walking, the experimenter reduced the speed of the treadmill to a stop over the course of 1 min and then remained stationary on the treadmill for a further minute, while heart rate was again recorded. A digital camera recorded the training from the side such that the experimenter's feet were in shot throughout.

### 2.2.3. Carrying experience—Slow condition

The procedure for the Slow condition was identical to the Fast condition except that the treadmill speed facilitated walking at 98 BPM (612 ms between steps), and the experimenter could hear a corresponding metronome recording of 98 BPM.

### 2.2.4. Caregiver measures

Following completion of infant testing, caregivers were asked to complete the following measures. Infants remained in the same room as the caregiver.

### 2.2.5. Caregiver spontaneous motor tempo measures

Caregivers took part in an abbreviated version of the infant drumming SMT task, where they were asked to sit within easy reach of the drum and drum consistently with one hand for 1 min, with a smooth gesture, and at a comfortable, regular rate. They also took part in a tapping task, where they tapped the surface of the drum with their index finger for 1 min, following the same instructions as when drumming.

### 2.2.6. Caregiver walking cadence measure

Caregivers were asked to step on to the stationary treadmill and were familiarized with the emergency stop. The caregiver then started the treadmill and the experimenter gradually increased and decreased the speed using a two-up two-down stair casing procedure (prompted: 'Is this rate better, or worse, than before?'), until the caregiver reported that they were walking at their most comfortable pace. The caregivers walked at this pace for 1 min. A video of the caregiver's footsteps was recorded in the profile. Walking cadence was measured in this way as there was no space for naturalistic overground walking (indoor or outdoor), and priority for the study was given to our primary, infant, measures. However, cadence is known to vary by context (e.g. [28]), and the stair casing procedure is unlikely to be as sensitive as a free-walking procedure. Such limitations are addressed in the discussion.

### 2.2.7. Questionnaire measure

Caregivers completed a questionnaire detailing infants' gross motor milestones and the amount of time their infant typically spends in different gross motor activities, including being carried in a sling.

### 2.2.8. Infant and caregiver anthropometric measurements

We took measurements of parent and infant height, leg length, arm length and weight. The experimenter took all measurements. Height was measured from the top of the head to the floor. Arm length was calculated by adding measurements from the spine to the shoulder to measurements from the shoulder to the wrist. Leg length was measured from the hipbone protrusion to the ankle. Adults were measured in a standing position. Infant arm and leg lengths were measured when standing (if able to hold themselves in a standing position), lying supine on the floor, or while being held by the caregiver, and height was always measured while lying supine on the floor. To calculate infant weight, the caregiver or experimenter stood on scales with and without the infant and the experimenter calculated the difference.

## 2.3. Apparatus

### 2.3.1. Measures of spontaneous motor tempo

Data were recorded using a Piezo contact microphone pickup fixed with adhesive tape to the underside of a 12-inch wood shell and natural skin head drum, attached to a height and angle adjustable mini-table. The pickup was connected to a Focusrite Scarlett 2i2 (American Music and Sound, MS, USA), a hardware interface connecting the microphone audio signal to the computer (MacBook Pro; Retina, 15-inch, Mid 2014). The Scarlett 2i2 was selected as the audio input and the sound recording was taken using Audacity®, v. 2.1.2 (2015). ScreenFlow (Telestream, Inc., CA, USA) was used to create a simultaneous screen capture of the Audacity recording and video footage of the infant/caregiver using the forward-facing built-in webcam.

### 2.3.2. Carrying experience

The walking experience was given on a Domyos Comfort Run treadmill, with 0% incline. Animations during the carrying experience were presented on a 12-inch video screen. Video recordings of the carrying session and parental cadence measure were conducted using a Logitech HD 1080p webcam positioned 1 m to the left of and facing the treadmill, allowing a profile view of the experimenter and infant.

### 2.3.3. Anthropometric measures

For all measures except height and weight, a standard soft textile tape measure was used. Caregiver height was measured against a line-measured wall. Infant height was measured by laying the infants on an infant height chart. Weights were taken on digital bathroom scales.

### 2.3.4. Heart rate data

Infant heart rate data were collected using a bipolar pediatric surface electrode (3M monitoring electrodes with micropore tape and solid gel) placed on the infant's back over the heart and recorded via a Myon 320 wireless EMG system, at a sampling rate of 4000 Hz.

## 2.4. Data processing

### 2.4.1. Measures of spontaneous motor tempo

For infants, the ScreenFlow video recordings of the drumming sessions were used to identify periods of drumming and determine the corresponding time point in the original Audacity sound file. The experimenter hand marked the onset of each hit (as defined by the first peak in the sound stream; figure 1a for example). For each 'bout' of drumming (i.e. series of hits), the time stamp of each hit onset was recorded, along with how many hits were in the bout, and whether the bout was produced by one hand drumming, both hands drumming simultaneously, or both hands in an alternating sequence. If data were so noisy that the onset of the drum hit was not distinguishable (i.e. because of wire noise, very low amplitude hitting, etc.), they were discarded. Each bout of drumming was considered separately, with a pause of more than 2 s between hits considered a break in drumming. To best match the adult literature on unimanual tapping, the following analyses were performed on the rate of unimanual hits, or on the first hand to strike during bimanual hits, with alternating sequences excluded.

Matlab (MATLAB R2015b, The MathWorks Inc., MA, USA) was used to calculate the inter-onset interval (IOI). The mean IOI was calculated for each participant and taken as a measure of SMT. The relative standard deviation (RSD; also known as the coefficient of variation—the ratio of the standard deviation to the mean, expressed as a percentage) of the IOI was also calculated for each participant and taken as a measure of regularity, i.e. a low RSD indicated more consistent drumming. To be included as a 'bout', infants had to perform four sequential hits with no more than a 2 s IOI between hits [6]. Infants who did not have at least one such 'bout' at pre- and post-test were excluded from further analyses. As a very high RSD would indicate that the infant was drumming with very little regularity and therefore the IOI would not be a good measure of SMT, two visual outliers with an

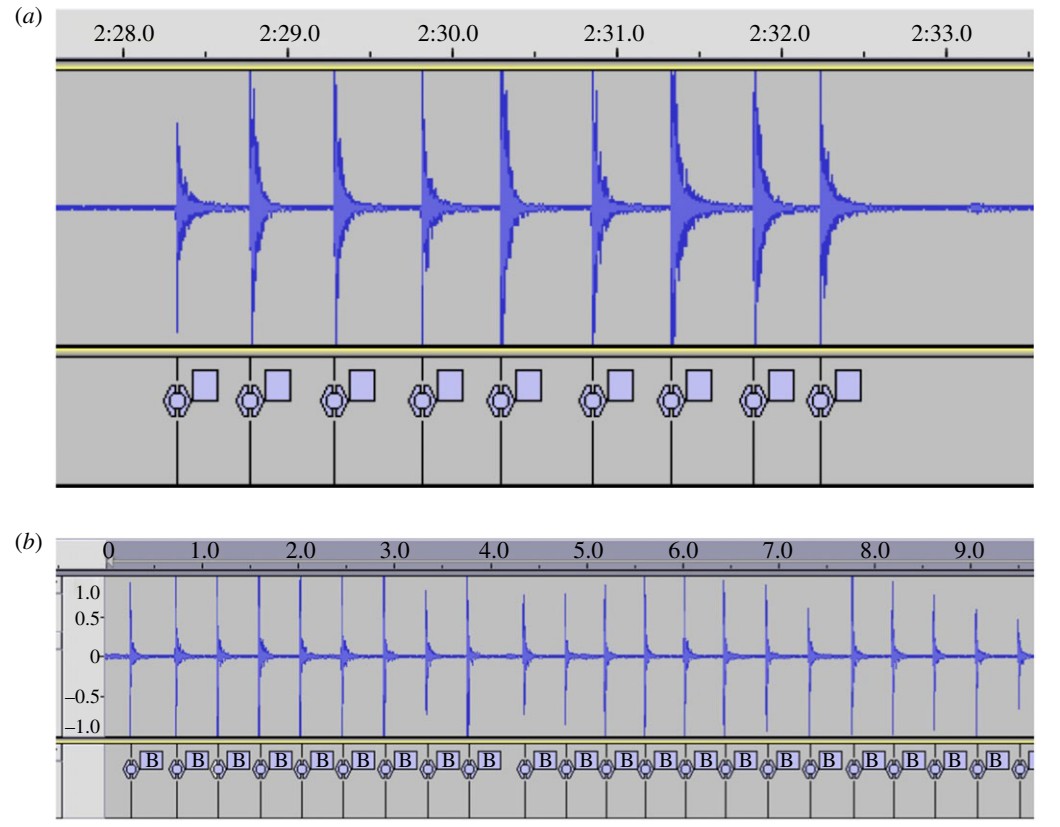

**Figure 1.** (*a*) Example of manually event marked sound stream of infant drumming. (*b*) Example of the event marking of an adult drumming, conducted with the Audacity® automatic beat finder function.

RSD of over 35% [6] were further excluded from the analysis of the intervention. This left a final sample of 27 infants with pre- and post-test data.

For caregiver drumming and caregiver tapping, the inbuilt Audacity 'Beat Finder' analysis tool was used to automatically detect and mark the onset of beats produced by the caregiver, by identifying each instance the signal went past a set decibel. This criterion was modified for each individual participant to account for individual variations in the strength of the hit/tap. The experimenter visually inspected the marked file and ensured all beats had been represented faithfully (figure 1*b* for example).

### 2.4.2. Caregiver cadence

The number of steps that each caregiver took in 1 min was coded from the profile view videotape. Steps per minute were translated to an IOI, giving milliseconds between steps as the independent variable, allowing easy comparison with the drumming and tapping data. The cadence of 11 caregivers was double coded, and the single-measure ICC for the IOI was 0.988, with a 95% confidence interval from 0.957 to 0.997 ($F_{10,10.7} = 170$, $p < 0.001$).

### 2.4.3. Heart rate

The infant EMG signal for the stationary period prior to walking and the stationary period immediately after walking were analysed using a custom-built ProEMG pipeline, marking the onset of each heartbeat. Data where the heartbeat was not evident due to wire noise or signal dropout were discarded. This left pre- and post-test data for 21 infants: 11 in the Fast condition and 10 in the Slow condition.

## 3. Results

Our primary hypothesis was that infant SMT would be influenced by experience of being carried at a novel rate; with infant SMT becoming faster from pre- to post-test if walked at the fast speed of 138 BPM and slower if walked at the slow speed of 98 BPM. This was confirmed with a repeated-

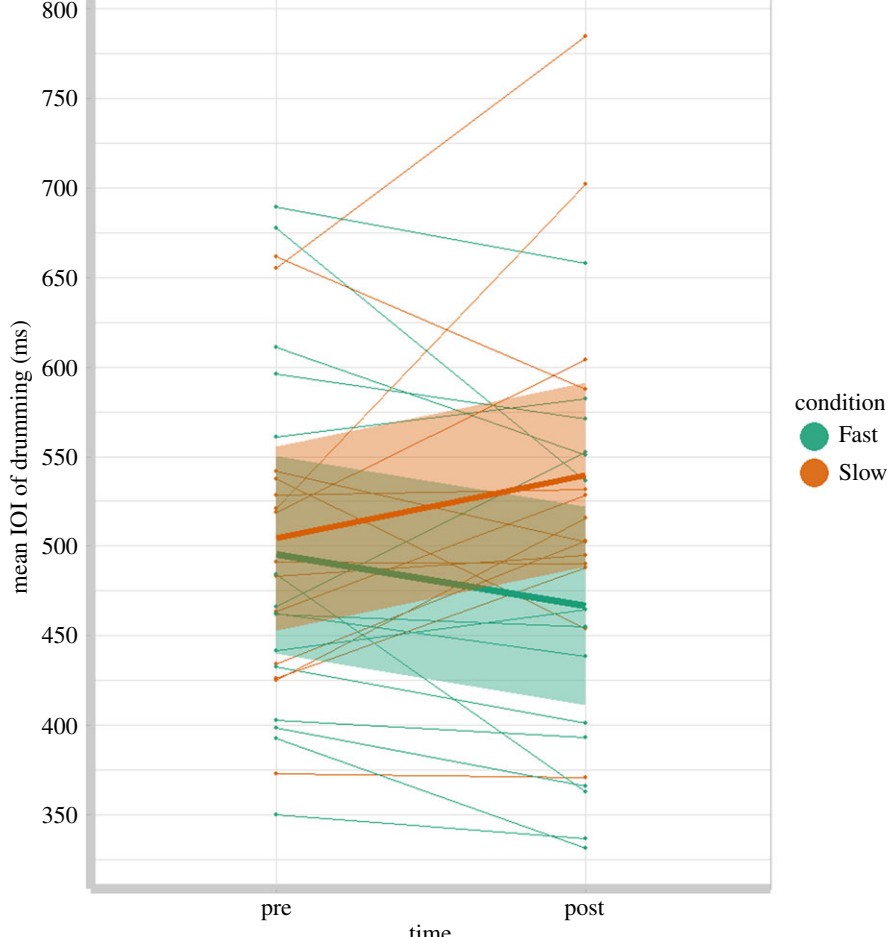

**Figure 2.** Graph to show the mean IOI of drumming at pre- and post-test for each infant, with regression lines (bold) and 95% confidence intervals (shaded) for the Fast and Slow walking conditions. Note: a faster SMT has a shorter IOI.

measures ANOVA with infant SMT as the dependent variable, time (pre-test or post-test) as a within-subject factor and condition (fast or slow) as a between-subjects factor, revealing no main effects of time ($F_{1,27} = 0.081$, $p = 0.778$) or condition ($F_{1,27} = 1.390$, $p = 0.249$) but a significant time × condition interaction ($F_{1,27} = 6.799$, $p = 0.015$, $\eta p^2 = 0.201$), such that infant SMT in the Fast condition became faster from pre- to post-test, and infant SMT in the Slow condition became slower from pre- to post-test (fast pre-test $M = 0.495$, s.e. = 0.025, 95% CI (0.445, 0.546); fast post-test $M = 0.467$, s.e. = 0.027, 95% CI (0.412, 0.521); slow pre-test $M = 0.504$, s.e. = 0.026, 95% CI (0.452, 0.557); slow post-test $M = 0.540$, s.e. = 0.028, 95% CI (0.484, 0.596); figure 2). One-tailed *post hoc* paired *t*-tests confirm that the changes in tempi were significant in both speed conditions (fast $t_{14} = 1.99$, $p = 0.0335$; slow $t_{13} = -1.752$, $p = 0.052$). At the individual level, 12 of the 15 infants in the Fast condition showed the expected pattern of a faster SMT at post-test than pre-test, and 9 of the 14 infants in the Slow condition showed a slower SMT at post-test than pre-test.

If a change in infant SMT were driven by higher arousal, or more physical exertion in the Fast condition, and lower arousal in the Slow condition, we would expect to see a faster heartbeat from pre- to post-test in the Fast condition and a slower heartbeat in the Slow condition. Figure 3 displays the mean heart rate IOI for infants in each condition. In the Fast condition, 5 of 11 infants showed an increase in heart rate, while 6 of 10 infants in the Slow condition showed an increase in heart rate. The pattern of change in heart rate is in the opposite direction to what would be predicted by a general change in arousal but is not statistically significant: a repeated-measures ANOVA with infant heart rate (IOI of heartbeats) as the dependent variable, time (pre-test or post-test) as a within-subject factor, and condition (Fast or Slow) as a between-subjects factor, confirms no significant effect of time ($F_{1,18} = 0.612$, $p = 0.442$) or condition ($F_{1,18} = 0.898$, $p = 0.356$), and no evidence of a time × condition interaction ($F_{1,18} = 1.585$, $p = 0.224$).

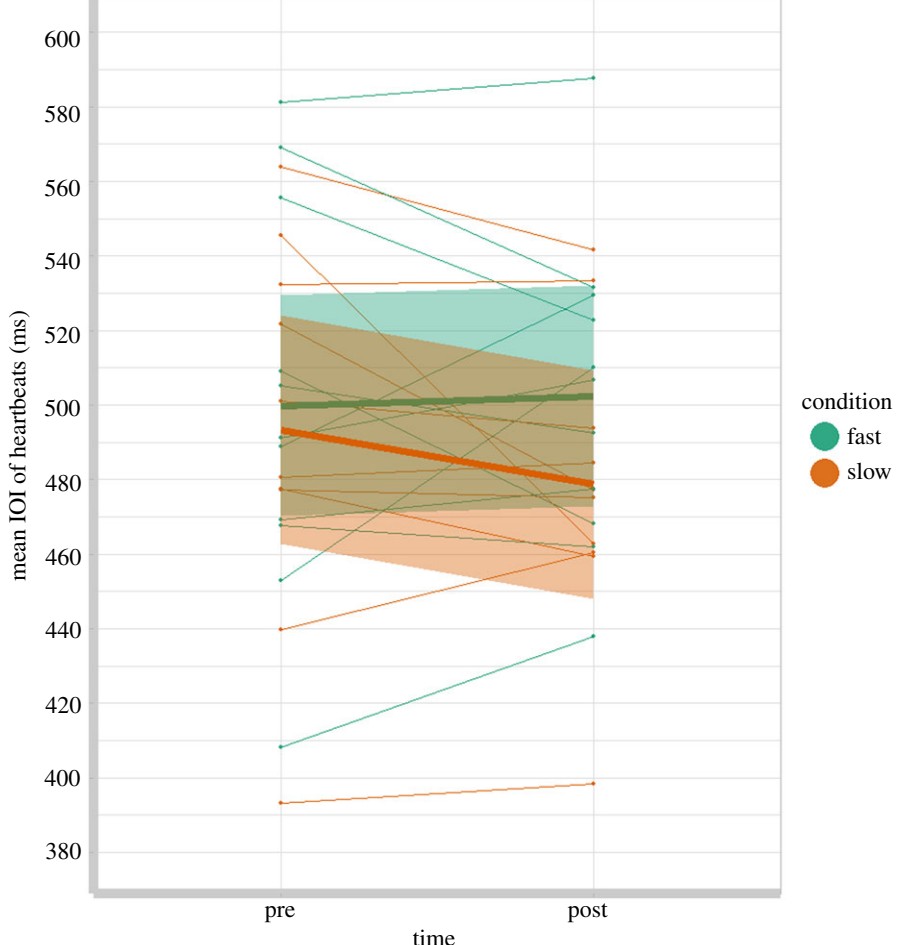

**Figure 3.** Graph to show heart rate at pre- and post-test for each infant, with regression lines (bold) and 95% confidence intervals (shaded) for the Fast and Slow walking conditions. Note: a faster heart rate has a shorter IOI.

In order to confirm that our data support the null hypothesis (no effect of carrying experience on infant heart rate) and are not the result of insufficient power, we used JASP (JASP Team 2017; v. 0.8.1.2) to calculate the Bayes factors for the interaction, using the default priors. It is assumed that $BF_{10} < 0.33$ provide good evidence to support the null [29,30]. The equivalent Bayesian repeated-measures ANOVA revealed evidence for the null (time × condition $BF_{10} = 0.263$), showing that there is over three times the evidence in our data that there is no effect of carrying rate on change in heart rate. We can therefore confirm that our SMT results are not explained by a change in arousal, as measured by heart rate.

## 3.1. Parent measures

In addition to our experimental manipulation, we were further interested in whether parents' rhythm as measured in the laboratory via drumming, tapping and crucially by walking cadence would predict infant SMT. A linear regression with infant SMT at pre-test as the dependent variable and caregiver drumming rate, tapping rate and stepping rate IOIs as predictors did not provide a significant model ($F_{3,28} = 1.255$, $p = 0.309$, $R^2 = 0.119$), and none of the predictors were significant (parent drum $\beta = 0.406$, $t_{28} = 1.897$, $p = 0.068$; parent tap $\beta = -0.173$, $t_{28} = -0.805$, $p = 0.421$; parent walking cadence $\beta = -0.025$, $t_{28} = -0.139$, $p = 0.891$).

Similarly, linear regressions reveal that both infant and parent anthropometrics fail to predict infant SMT, and that parental anthropometrics fail to predict parent drumming and tapping. Table 1 displays the standardized coefficients, $t$- and $p$-values for all body measurements, both infant and parent, when predicting the dependent variable of infant pre-test SMT (infant measurements, $F_{4,30} = 1.840$, $p = 0.147$, $R^2 = 0.197$; parent measurements $F_{4,27} = 0.220$, $p = 0.925$, $R^2 = 0.032$); and for parent measurements

predicting parent drumming ($F_{4,36} = 0.613$, $p = 0.656$, $R^2 = 0.064$) and parent tapping ($F_{4,36} = 1.491$, $p = 0.225$, $R^2 = 0.142$).

However, we do see a significant contribution of parent body size to parent walking cadence ($F_{4,30} = 3.832$, $p = 0.012$, $R^2 = 0.338$). This is driven by a highly significant contribution of parent weight, such that heavier adults have a slower walking cadence than lighter adults (parent weight $\beta = 0.500$, $t_{30} = 2.804$, $p = 0.009$). Parent weight is positively correlated with the other parent anthropometrics (all $p < 0.01$), and all parent anthropometrics are positively correlated with walking cadence, such that larger measurements correlate with slower walking, though only weight and arm length reach significance (for correlations, see table 2). As the anthropometrics were highly correlated, it is possible that overall stature is the best predictor of walking cadence. We therefore calculated a height × weight composite score (stature) and entered this as a predictor for parental cadence in a simple linear regression. The new parent stature variable predicts walking cadence ($F_{1,34} = 10.092$, $p = 0.003$, $R^2 = 0.229$).

The current data fail to replicate that parental height predicts infant SMT [6]. This is likely due to the smaller sample size in the current study ($N = 32$) than in the former study ($N = 62$). A Bayesian linear regression with the same factorial design as in Rocha *et al.* [6] confirms that this model has inconclusive Bayes factors (around 1) for all predictors (parent height BF10 = 0.621), showing that there is no evidence for an absence of an effect. Incorporating the ten-month-old data from the current study with the sample collected in Rocha *et al.* [6], the previous finding of a significant model and parental height predicting infant SMT stands ($F_{6,85} = 3.576$, $p = 0.003$, $R^2 = 0.202$; parent height $\beta = 0.364$, $t_{85} = 2.509$, $p = 0.014$).

## 3.2. Infant motor activity

Finally, we were interested in whether the rate or regularity of infant SMT was related to the types of experience of own locomotion and caregiver locomotion in which the infant participates, during her daily life. Two linear regressions with infant SMT and the RSD of infant SMT as dependent variables, and whether parents reported infants were carried for more than 30 min a day, could crawl or could cruise entered as predictors and revealed that although motor activity did not predict the rate at which infants drummed (infant SMT, $F_{3,31} = 0.887$, $p = 0.459$, $R^2 = 0.079$), we find a significant model for the contribution of these activities to the variability in infant data (infant pre-test RSD, $F_{3,31} = 4.037$ $p = 0.016$, $R^2 = 0.281$), explaining 28% of the variance in the data. This result is driven by a highly significant effect of whether infants could cruise on infant RSD, such that infants who were cruising at the time of testing were more variable in their drumming (cruising $\beta = 0.532$, $t_{31} = 3.349$, $p = 0.002$). Full results are displayed in table 3.

## 4. Discussion

We predicted that the experience of being carried at either a faster (138 BPM) or slower (98 BPM) than average walking pace would directly influence the SMT of ten-month-old non-walking infants. While performance between the two groups did not differ at pre-test, following 10 min of being carried at a novel pace, infants in the Fast group showed a faster SMT at post-test than infants in the Slow group, with a mean rate of change across conditions of 20–30 ms. We are thus the first to provide direct evidence that carrying infants can bias the rhythms that they spontaneously produce. Further, monitoring of infants' heart rate before and after the walking training revealed no change in arousal from pre- to post-test, suggesting that the impact on SMT may be specific to rhythm, and unlikely to be the result of a general state change.

We took additional correlational measures of infant and parent body size, and parent rhythm, including parent cadence. Parental body size predicted parental walking cadence, partially corroborating the interpretation of Rocha *et al.* [6], that parental body size predicts infant SMT as parent body size sets the tempo of a parent walking. However, it was not the hypothesized height measurement that drove this result. Although the parental anthropometrics were all highly correlated, when measurements were entered into a multiple regression, weight was the strongest predictor, with arm length also marginally significant. That arm length was a predictor may reflect that it is a composite score, consisting of breadth (spine to shoulder) plus length (shoulder to wrist). When measuring cadence, indicators of overall stature may be more important than length measurements alone. Prior research on body size and walking cadence do show an impact of both weight and height, and both are normally used to standardize gait measurements (e.g. [31]). This interpretation is

**Table 1.** Linear regression coefficients for effects of infant and parent anthropometrics on measures of infant and parent SMT.

| | infant drum | | | parent drum | | | parent tap | | | parent walk | | |
|---|---|---|---|---|---|---|---|---|---|---|---|---|
| | β | t | p | β | t | p | β | t | p | **β** | t | p |
| infant arm | 0.255 | 1.388 | 0.175 | / | / | / | / | / | / | / | / | / |
| infant leg | 0.142 | 0.752 | 0.458 | / | / | / | / | / | / | / | / | / |
| infant height | 0.284 | 1.313 | 0.199 | / | / | / | / | / | / | / | / | / |
| infant weight | −0.137 | −0.634 | 0.531 | / | / | / | / | / | / | / | / | / |
| parent arm | −0.133 | −0.486 | 0.631 | −0.086 | −0.386 | 0.702 | 0.034 | 0.159 | 0.875 | 0.367 | 1.756 | 0.089 |
| parent leg | −0.019 | −0.060 | 0.953 | −0.286 | −0.990 | 0.329 | −0.253 | −0.916 | 0.366 | −0.100 | −0.337 | 0.738 |
| parent height | −0.025 | −0.083 | 0.935 | 0.137 | 0.483 | 0.632 | −0.196 | 0.722 | 0.475 | −0.244 | −0.837 | 0.409 |
| parent weight | 0.097 | 0.426 | 0.674 | 0.157 | 0.824 | 0.415 | 0.268 | 1.466 | 0.151 | 0.500 | 2.804 | 0.009* |

**Table 2.** Correlations between parent anthropometrics.

|  | parent arm | parent leg | parent height | parent weight | walking cadence |
|---|---|---|---|---|---|
| parent arm |  | 0.642*** | 0.581*** | 0.454** | 0.319^ |
| parent leg |  |  | 0.788*** | 0.380* | 0.137 |
| parent height |  |  |  | 0.455** | 0.146 |
| parent weight |  |  |  |  | 0.478** |

$*p > 0.05$, $**p > 0.01$, $***p > 0.001$, $^{\wedge}p = 0.051$.

**Table 3.** Linear regression coefficients for effects of motoric experience on infant SMT and infant RSD.

|  | infant SMT | | | infant RSD | | |
|---|---|---|---|---|---|---|
|  | $\beta$ | $t$ | $p$ | $\beta$ | $t$ | $p$ |
| crawl | −0.199 | −1.120 | 0.271 | −0.231 | −1.472 | 0.151 |
| cruise | 0.234 | 1.305 | 0.201 | 0.532 | 3.349 | 0.002 |
| sling use | −0.053 | −0.305 | 0.763 | −0.013 | −0.086 | 0.932 |

supported by the significant predictive power of a composite parent stature measure. While we fail to replicate the prior finding that parent height predicts infant SMT [6], this is likely due to a lack of statistical power, as Bayesian analyses reveal that we are not finding evidence that support the null, but rather the data are inconclusive.

While parent body size predicted parental cadence, we did not find an impact of parental cadence on parent's own SMT as measured by drumming or tapping, or on infant SMT as measured through drumming at pre-test. As such, we are unable to claim from our correlational measures that infant SMT is related to the experience of locomotion at their caregiver's walking cadence. Our measure of caregiver cadence is a limitation of the study. Treadmill walking is known to be different from overground walking (e.g. [32]), and in our case, forced a non-continuous dependent variable, as there were limited speed options to walk at. For health and safety reasons, the caregiver was unable to carry their own child while on the treadmill, and caregiver motivation to be precise may have been low, especially if their infant was beginning to tire or fuss. That we do not find relationships within our adult measures may therefore reflect our suboptimal procedures (that were a direct consequence of our bespoke infant testing environment). Further, while the caregiver tested was the primary caregiver at time of the appointment, the recent introduction of equal maternity and paternity rights meant that we saw a mix of mothers and fathers. Though there was no difference in results, if fathers were removed from the analyses reported, we did not collect specific data on time spent with different adults, and multiple caregiver families may weaken the effect we predicted to see. Future research should measure each caregiver with significant responsibility for the child. However, it is important to note that in the controlled laboratory environment our experimental manipulation worked, with the novel experience changing infant SMT in the hypothesized directions, after only a 10 min intervention. The limitations we note here can be rectified in future work by recording parents' natural behaviour, using accelerometers worn by the infant and caregiver. This kind of rich longitudinal time-series data will be critical for a sensitive analysis exploring how the natural rhythms experienced in daily life impact infant SMT.

We further find that infants who were cruising (walking with assistance) at the point of testing were more variable in their drumming than infants who were not cruising. Increased vestibular experience may benefit cognitive and motor functions [33–36], likely through increased variability of experience [37]. Our findings are in line with Thelen's documentation of rigid, rhythmic stereotypies disappearing after the onset of more mature, volitional action; flexible and complex behaviour supersedes the regularity of initial motor outputs [38]. While across the lifespan more variability in SMT may be viewed as a less mature rhythmic response, it is possible that in the first year of life, this less rigid performance may reflect the beginnings of greater motor control gained from the infants' own, diverse, locomotive experience.

The current findings show that we can alter infant rhythm production with novel experience of locomotion, but future longitudinal work is necessary to understand the extent to which naturalistic experience of locomotion, from prenatal experience, to postnatal infant carrying, to the establishment of self-locomotion, impact our musicality. Outstanding questions include how long the effects of locomotive experience last, whether such effects are universal, and whether such experience is necessary for typical rhythm development.

Through a pre-test, training, post-test design, we successfully manipulated infant SMT with a 10 min novel walking pace carrying intervention. This is the first direct evidence that carrying infants can change the rhythms that infants naturally produce. Our results suggest that an experience generally regarded as 'passive' on behalf of the infant may be shaping their earliest musical tendencies.

Ethics. The research complied with APA ethical standards and the Declaration of Helsinki, and received ethical approval from the Department of Psychological Sciences, Birkbeck University of London. All caregivers gave written, informed consent concerning the experimental procedure for themselves and their infants.

Data accessibility. Data are provided as electronic supplementary material [39].

Authors' contributions. S.R., V.S. and D.M. conceived the study. S.R. collected and analysed the data. S.R., V.S. and D.M. wrote the manuscript. All authors gave final approval for publication and agree to be held accountable for the work performed therein.

Competing interests. We declare we have no competing interests.

Funding. We received funding for this study from Economic and Social Research Council.

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

of infant carrying impacts infant spontaneous
motor tempo. Figshare.

