## [Peer Review File · Royal Society Open Science]

Review History

RSOS-210608.R0 (Original submission)

Review form: Reviewer 1 (Matz Larsson)

Is the manuscript scientifically sound in its present form?

Yes

Are the interpretations and conclusions justified by the results?

Yes

Is the language acceptable?

Yes

Do you have any ethical concerns with this paper?

No

Have you any concerns about statistical analyses in this paper?

No

Recommendation?

Accept with minor revision (please list in comments)

Comments to the Author(s)

This study explores if infant SMT is influenced by the rate of locomotion infants experience when being carried. Ten-month-old, non-walking infants were tested using a free drumming procedure before and after being carried at a slower (98 BPM) or faster (138 BPM) than average human walking tempo. The study suggests that infant SMT is influenced by the tempo at which they were carried. The study is interesting and provides new data in a scarcely investigated field. The number of investigated children is moderate but nevertheless the hypothesis of a link between carrying type and SMT of children. The hypothesis that antropometrics of parents could be associated with SMT of the child was not supported. The authors suggest that it may be due to the low number and accordingly lack of statistical power. That may be the case but I will discuss another possible reason below.

Line 482 "The tempo of most music and human locomotion coincide (Mishima, 1965; Ayres, 1973; Fraisse, 1982; Styns et al., 2007; Trainor, 2007). Here might be mentioned two more recent papers that suggest a natural selection hypothesis behind this coincidence suggesting a that if locomotion sound are produced in pace by walking humans it will result in less masking problems :

Larsson, M. (2014). Self-generated sounds of locomotion and ventilation and the evolution of human rhythmic abilities. *Animal Cognition*, 17, 1-14. doi:10.1007/s10071-013-0678-z

Review form: Reviewer 2

Is the manuscript scientifically sound in its present form?

Yes

Are the interpretations and conclusions justified by the results?

Yes

Is the language acceptable?

Yes

Do you have any ethical concerns with this paper?

No

Have you any concerns about statistical analyses in this paper?

No

Recommendation?

Accept with minor revision (please list in comments)

Comments to the Author(s)

A few additional notes about typos/points of clarification:

It is not clear what the black lines are in Figures 2 and 3.

Sentence lines 469-474 is long and hard to parse.

Double-negative lines 537-538

The introduction could be re-organized to flow better. In places the connection between adjacent sentences was not clear and the ideas felt jumbled. Likewise, some of the paragraphs seemed to be in the wrong order.

Line 642 delete "method"

The text about the effect of vestibular stimulation on arousal and specifically about heart rate post-test (lines 752-769) is a bit confusing. In the first paragraph there is a pointer to a figure but no statement of the results. The next paragraph then starts with "Though these results..." as if the results had been given. Only then is there a statement that the effect was not significant.

Decision letter (RSOS-210608.R0)

Dear Dr Rocha

On behalf of the Editors, we are pleased to inform you that your Manuscript RSOS-210608 "Rate of infant carrying impacts infant spontaneous motor tempo" has been accepted for publication in Royal Society Open Science subject to minor revision in accordance with the referees' reports. Please find the referees' comments along with any feedback from the Editors below my signature.

Please submit your revised manuscript and required files (see below) no later than 7 days from today's (ie 02-Aug-2021) date. Note: the ScholarOne system will 'lock' if submission of the revision is attempted 7 or more days after the deadline. If you do not think you will be able to meet this deadline please contact the editorial office immediately.

on behalf of Dr Teodora Gliga (Associate Editor) and Essi Viding (Subject Editor)
openscience@royalsociety.org

Associate Editor Comments to Author (Dr Teodora Gliga):

As you will see both reviewers found your study particularly original, the methods rigorous and the evidence compelling. They both also make suggestions that will improve the clarity of the paper and the depth of the discussion. I therefore ask you to revise the manuscript incorporating changes in response to these comments. Pay particular attention to the "flow" of the argument and data reporting. For example, some of your analysis are introduced before being justified (move the text that justifies your investigation of heart rate from the result section to the introduction) or are never justified - the analysis of rhythm variability, albeit interesting, comes a little out of the blue. I also agree that a more in depth discussion of parental sex is needed as the contribution of only one parent may be pre-natal and both parents may contribute carrying experience albeit in different proportions.

Associate Editor: 2

Comments to the Author:

(There are no comments.)

Reviewer comments to Author:

Reviewer: 1

Comments to the Author(s)

This study explores if infant SMT is influenced by the rate of locomotion infants experience when being carried. Ten-month-old, non-walking infants were tested using a free drumming procedure before and after being carried at a slower (98 BPM) or faster (138 BPM) than average human walking tempo. The study suggests that infant SMT is influenced by the tempo at which they were carried. The study is interesting and provides new data in a scarcely investigated field. The number of investigated children is moderate but nevertheless the hypothesis of a link between carrying type and SMT of children. The hypothesis that antropometrics of parents could be associated with SMT of the child was not supported. The authors suggest that it may be due to the low number and accordingly lack of statistical power. That may be the case but I will discuss another possible reason below.

Line 482 "The tempo of most music and human locomotion coincide (Mishima, 1965; Ayres, 1973; Fraisse, 1982; Styns et al., 2007; Trainor, 2007). Here might be mentioned two more recent papers that suggest a natural selection hypothesis behind this coincidence suggesting a that if locomotion sound are produced in pace by walking humans it will result in less masking problems:

Larsson, M. (2014). Self-generated sounds of locomotion and ventilation and the evolution of human rhythmic abilities.

Animal Cognition, 17, 1-14. doi:10.1007/s10071-013-0678-z

Reviewer: 2

Comments to the Author(s)

A few additional notes about typos/ points of clarification:

It is not clear what the black lines are in Figures 2 and 3.

Sentence lines 469-474 is long and hard to parse.

Double-negative lines 537-538

The introduction could be re-organized to flow better. In places the connection between adjacent sentences was not clear and the ideas felt jumbled. Likewise, some of the paragraphs seemed to be in the wrong order.

Line 642 delete “method”

The text about the effect of vestibular stimulation on arousal and specifically about heart rate post-test (lines 752-769) is a bit confusing. In the first paragraph there is a pointer to a figure but no statement of the results. The next paragraph then starts with “Though these results...” as if the results had been given. Only then is there a statement that the effect was not significant.

===PREPARING YOUR MANUSCRIPT===

one version identifying all the changes that have been made (for instance, in coloured highlight, in bold text, or tracked changes);
a 'clean' version of the new manuscript that incorporates the changes made, but does not highlight them. This version will be used for typesetting.

===PREPARING YOUR REVISION IN SCHOLARONE===

<https://royalsociety.org/journals/authors/author-guidelines/#supplementary-material> to include a suitable title and informative caption. An example of appropriate titling and captioning may be found at https://figshare.com/articles/Table_S2_from_Is_there_a_trade-off_between_peak_performance_and_performance_breadth_across_temperatures_for_aerobic_scops_in_teleost_fishes_/3843624.

Author's Response to Decision Letter for (RSOS-210608.R0)

See Appendix A.

Decision letter (RSOS-210608.R1)

Dear Dr Rocha,

I am pleased to inform you that your manuscript entitled "Rate of infant carrying impacts infant spontaneous motor tempo" is now accepted for publication in Royal Society Open Science.

on behalf of Dr Teodora Gliga (Associate Editor) and Essi Viding (Subject Editor)
openscience@royalsociety.org

Appendix A

Response to reviews for RSOS-210608. Author replies are in italics.

Associate Editor Comments to Author (Dr Teodora Gliga):

As you will see both reviewers found your study particularly original, the methods rigorous and the evidence compelling. They both also make suggestions that will improve the clarity of the paper and the depth of the discussion. I therefore ask you to revise the manuscript incorporating changes in response to these comments. Pay particular attention to the "flow" of the argument and data reporting. For example, some of your analysis are introduced before being justified (move the text that justifies your investigation of heart rate from the result section to the introduction) or are never justified - the analysis of rhythm variability, albeit interesting, comes a little out of the blue. I also agree that a more in depth discussion of parental sex is needed as the contribution of only one parent may be pre-natal and both parents may contribute carrying experience albeit in different proportions.

Thank you for the insightful comments on our manuscript. We hope that we have addressed all of the points raised by yourself and the reviewers, detailed below, and believe the flow of the manuscript is now much improved. All of the analyses are now justified prior to their presentation, with the results clearly stated in the appropriate location.

Regarding the contribution of multiple parents, we have more explicitly stated that measurement of all caregivers, and more nuanced measurement at that, is necessary for bolder claims on the impact of pre- and post-natal carrying experience on rhythm. However, as we state in response to Reviewer 1 below, it is of utmost importance to the current thesis that even short periods of carrying experience impact infant rhythm production; this is the crux of our intervention. This is consistent with prior established findings of short periods of similar experience impacting infant rhythm perception, as cited within the manuscript. Of course the relative strength of such experiences in daily life, and through highly divergent periods of development, will vary hugely. We now acknowledge this more directly in the discussion and hope that this provokes much needed future work in this area. The following short penultimate paragraph has been added, Lines 510-516: 'The current findings show that we can alter infant rhythm production with novel experience of locomotion, but future work is necessary to understand the extent to which naturalistic experience of locomotion, from prenatal experience, to postnatal infant carrying, to the establishment of self-locomotion, impact our musicality. Outstanding questions include how long the effects of locomotive experience last, whether such effects are universal, and whether such experience is necessary for typical rhythm development.'

Associate Editor: 2

Comments to the Author:

(There are no comments.)

Reviewer comments to Author:

Reviewer: 1
Comments to the Author(s)

This study explores if infant SMT is influenced by the rate of locomotion infants experience when being carried. Ten-month-old, non-walking infants were tested using a free drumming procedure before and after being carried at a slower (98 BPM) or faster (138 BPM) than average human walking tempo. The study suggests that infant SMT is influenced by the tempo at which they were carried. The study is interesting and provides new data in a scarcely investigated field. The number of investigated children is moderate but nevertheless supports the hypothesis of a link between carrying type and SMT of children. On the other hand, the hypothesis that antropometrics of parents could be associated with SMT of the child was not supported. The authors suggest that it may be due to the low number and accordingly lack of statistical power. I agree that may be the case but I will discuss another possible reason below.

In the abstract the authors advocates “that being carried during caregiver locomotion is a predominant experience for infants throughout the first years of life, and as a source of regular, vestibular, information, may at least partially form the basis of their sense of rhythm”. I certainly agree with that but I think the prenatal period ought to be discussed more, since that period, at least the last part of it will serve as a source of regular, vestibular, information.

Line 482 "The tempo of most music and human locomotion coincide (Mishima, 1965; Ayres, 1973; Fraise, 1982; Styns et al., 2007; Trainor, 2007). Here might be mentioned a more recent paper that suggest a natural selection hypothesis behind this coincidence of tempi (locomotion sound produced in pace by walking humans is likely to result in less masking problems and improved hearing in dangerous terrain):

Larsson, M. (2014). Self-generated sounds of locomotion and ventilation and the evolution of human rhythmic abilities. *Animal Cognition*, 17, 1–14.
doi:10.1007/s10071-013-0678-s

We too find the idea of synchronous locomotion producing a cleaner and more predictable signal very compelling. We have added the Larsson (2014) reference as suggested.

Line 500 Richard Parncutt have suggested a link between maternal footfall and music long before Teie. Here are some references:

Parncutt, R. (1987). The perception of pulse in musical rhythm. In A. Gabrielsson (Ed.), *Action and perception in rhythm and music* (pp. 127–138). Stockholm, Sweden: Royal Swedish Academy of Music, Publication No 55.

Parncutt, R. (1989). *Harmony: A psychoacoustical approach*. Berlin, Germany: Springer.

Parncutt, R. (1993). Prenatal experience and the origins of music. In T. Blum (Ed.), *Prenatal perception, learning, and bonding* (pp. 253–277). Berlin, Germany: Leonard.

Thank you for introducing the work of Richard Parncutt, which we have not come across before. We have now preceded the discussion of Teie with a brief summary of Parncutt's earlier work, see Line 70-72: 'Parncutt (1989) suggests that prenatal conditioning of the contingency of these two signals via locomotion is responsible for the link between auditory rhythm and body movement.'

Both Parncutt and Teie discusses footfalls but in addition several other putative sound that might be heard in the womb and eventually influence musical interest such as heart beat, blood-vessels, breathing sound, guts... However, a recent paper Larsson, M., Richter, J., & Ravignani, A. (2019). Bipedal Steps in the Development of Rhythmic Behavior in Humans. *Music & Science*, 2, 1-14 argues that maternal footfalls during pregnancy provides a multimodal rhythmic experience and hence is likely to be more influential than other sound heard in the womb in regards of the development of musical interest and rhythm. It also argues why bipedal walking during pregnancy may be part of the explanation to why other primates by large lack musical abilities. I just cite an example of the arguments:

"If the sound and vibrations produced by footfalls of a walking mother are transmitted to the fetus in coordination with the cadence of the motion, a connection between isochronous sound and rhythmical movement may be developed. Rhythmical sounds of the human mother locomotion differ substantially from that of nonhuman primates, while the maternal heartbeat heard is likely to have a similar isochronous character across primates, suggesting a relatively more influential role of footfall in the development of rhythmic/musical abilities in humans."

*Thank you, this is a very important point and one with which we wholeheartedly agree. We have now included the crux of this idea to our introduction, see Line 74-77: 'Critically, the **multimodal** nature of locomotive experience (as compared, for example, to fetal auditory perception of maternal heart rate or respiration) distinguishes locomotion from other rhythmic prenatal experiences (Larsson, Richter & Ravignani, 2019).'*

This article may also be of interest for the authors: Mariette et al 2021 Acoustic developmental programming: a mechanistic and evolutionary framework Line 516-9

In a recent study Rocha et al. (2021) developed a simple free drumming task that measured the SMT of infants across the first years of life, and found that infant SMT was predicted by parental body size, but not infant's own body size. Specifically, parent height predicted infant SMT such that infants with taller parents showed a slower SMT than infants with shorter parents.

This is interesting data. However, that study as well as the present lack data about the sex of caregiver, i.e. if it was the father or the mother that were present during the experiment. It would be interesting and relevant with data about parental sex to compare if it is a stronger association between child SMT and mothers size compared with the father.

The finding Rocha et al. (2021) that big fathers (also) were associated with an increased likelihood of having children that produces slow SMT may be an indirect association. Tall mothers are more likely to have tall male partners.

Thus, the finding about tall parents may not be casually linked with parents carrying the child – it may be a result of maternal walking during pregnancy. The authors should discuss that.

In future studies I suggest that researchers in this field include antropometric data of both parents. That could solve the question - is it rhythmic experience before or after birth that is (most) important? It may be both in combination of course.

We thoroughly agree that data from both parents would be ideal, though this was not possible in this experiment. We have now explicitly added to the discussion Line 489: ‘Future research should measure each caregiver with significant responsibility for the child.’ The suggestion that the impact of fathers may be indirect via mate choice is very interesting, and certainly something we could look at in future work with both parents. However, it is also important to note that the results presented from the experimental manipulation of carrying suggest that a relatively short period of experience with a novel caregiver (the experimenter), walking at a novel pace, impacted infant SMT. Of course, this is not to depreciate the importance of prenatal experience on rhythmic preference – but that question is a much larger one than that which we attempt to answer in this empirical paper. We have further stressed this with a new penultimate paragraph, Lines 510-516: ‘The current findings show that we can alter infant rhythm production with novel experience of locomotion, but future work is necessary to understand the extent to which naturalistic experience of locomotion, from prenatal experience, to postnatal infant carrying, to the establishment of self-locomotion, impact our musicality. Outstanding questions include how long the effects of locomotive experience last, whether such effects are universal, and whether such experience is necessary for typical rhythm development.’

"The current data fail to replicate that parental height predicts infant SMT (Rocha et al., 2021). This is likely due to the smaller sample size in the current study (N = 32) " I think that the sex of the carrying parent would be relevant information. How many were males/ females?

Five of the caregivers tested were male. This has now been added to the methods section, describing the participants: Line 169: ‘Five of the participating caregivers were male.’ As stated in the discussion, removing the fathers from the analyses did not change our results.

I suggest the manuscript should be accepted with minor revision.

Thank you very much for your constructive comments.

Reviewer: 2

Comments to the Author(s)

Overall, this is a fine paper. It examines an interesting issue that to my knowledge has not received much attention in the literature and makes a contribution. The methods are inventive and sound. I think the interpretations are largely justified. The one potential issue is that while the hypotheses is

stated at the individual level, the results are all presented at the group level. That is, the finding is that the two groups do not differ in SMT at pre-test but do differ at post-test, with the infants who were carried at a fast pace showing faster SMT and those carried at a slow pace showing slower SMT. Would it be possible to also look at this at the individual level? I think the hypothesis would predict that at the individual level infants carried at a fast pace should show a positive score if pre SMT was subtracted from post (and vice versa for those in the slow condition). It might be that the sample size is a bit small to show this with infant variability. Also, Figure 2 obscures the ability to look at this at the individual level. If individual lines were shown rather than the overall distribution we could see if the green lines for the fast have a positive slope while the orange lines for the slow condition have a negative slope.

Thank you for your comments. We agree with the prediction that infants would show a faster SMT at post-test than pre-test in the Fast condition, and vice versa for the slow condition. We carried out your suggestion of subtracting the post-test SMT from the pre-test SMT and found that 12/15 infants in the Fast condition showed a positive value, and 9/14 infants in the Slow condition showed a negative value. These numbers are now reported on line 351: 'At the individual level, 12 of the 15 infants in the Fast condition showed the expected pattern of a faster SMT at post-test than pre-test, and 9 of the 14 infants in the Slow condition showed a slower SMT at post-test than pre-test.'

We have additionally adapted Figure 2 to display the individual slopes from pre- to post-test for each infant. Further, to clarify the figure, rather than displaying the mean and standard error for each group at each time point, we now show a linear regression line with 95% CI for each group. Originally the figure additionally showed the distribution of the data via a split violin plot, but this became too cluttered on the new figure, and was therefore removed.

My only other comment at this level is that it is not clear why a different method was used to extract the drumming data for the infants compared to the adults. I would assume this is related to the force of the infants, but an explicit statement about why an automatic (and presumably less prone to bias) method was not used for the infants. Also, there is no indication of reliability coding for infant drumming or caregiver cadence.

Indeed, the automatic procedure worked well for adults as they were consistent in the force used – some may have drummed loudly and others quietly, but a threshold could be set for each adult that distinguished a beat from any other noise. The primary issue for automatic coding was that infants were much more variable in their force, sometimes hitting the drum very hard, and sometimes very gently, and potentially interspersed with other actions, such as scratching. It was not feasible to set one level for each baby that captured their drum hits with anywhere near the same accuracy as a human coder. We do not report a reliability measure here, but the reliability of the identical procedure with the same primary coder was reported in Rocha et al. (2021): 'An independent researcher blind to the aims of the study double-coded the video data for 30 infants. The single-measure ICC for the Inter-

Onset-Interval (IOI) was .924, with a 95% confidence interval from .790 to .968, ($F(29, 29) = 33.353, p < .001$).

The cadence of eleven caregivers was double coded by a naïve coder, and is now reported on line 328: 'The cadence of 11 caregivers was double coded, and the single-measure ICC for the IOI was .988, with a 95% confidence interval from .957 to .997 ($F(10, 10.7) = 170, p < .001$).

re the data submitted: It would be very helpful if a text file could be added with definitions of all the variables. Most are transparent and can be matched up to the manuscript text, but noting what "cleaned" means, for example, would prevent potential miscommunication or misunderstanding.

Thank you, a 'data dictionary' now accompanies the open dataset. In this case 'Cleaned' simply meant that it met inclusion criteria as defined in the methods section (i.e. this is not the raw data), therefore we have now removed the preface 'cleaned' to avoid confusion.

A few additional notes about typos/points of clarification:
It is not clear what the black lines are in Figures 2 and 3.

The figures have been altered and relabelled. The figure captions now fully describe all elements of the plots.

Sentence lines 469-474 is long and hard to parse.

These sentences have been altered and shortened in respect to this comment and with the general reflow of the introduction.

Double-negative lines 537-538

Clarified, and moved to Line 177: 'The Fast and Slow rates reflect the extreme values of the normal range of cadence in free-speed walking for females aged 18-49 years (Whittle, 1990).'

The introduction could be re-organized to flow better. In places the connection between adjacent sentences was not clear and the ideas felt jumbled. Likewise, some of the paragraphs seemed to be in the wrong order.

The introduction has been heavily restructured to flow better, and the connections between ideas have been clarified. Hopefully this is a much clearer manuscript.

Line 642 delete "method"

Deleted. Thanks.

The text about the effect of vestibular stimulation on arousal and specifically about heart rate post-test (lines 752-769) is a bit confusing. In the first

paragraph there is a pointer to a figure but no statement of the results. The next paragraph then starts with “Though these results...” as if the results had been given. Only then is there a statement that the effect was not significant.

*This section was overly convoluted for the point it needed to make and has been cut considerably. The literature on infant HR has been removed as it was not necessary, and the results have been clarified and better match the presentation of the drumming results. Line 355 - 368: ‘If change in infant SMT were driven by higher arousal, or more physical exertion in the Fast condition, and lower arousal in the Slow condition, we would expect to see a faster heartbeat from pre- to post-test in the Fast condition and a slower heartbeat in the Slow condition. Figure 3 displays the mean heart rate IOI for infants in each condition. In the Fast condition five of 11 infants showed an increase in heart rate, and four of 10 infants in the Slow condition showed a decrease in heart rate. The pattern of change in heart rate is in the opposite direction to what would be predicted by a general change in arousal but is not statistically significant: A repeated measures ANOVA with infant heart rate (IOI of heartbeats) as the dependent variable, Time (Pre-Test or Post-Test) as a within subject factor, and Condition (Fast or Slow) as a between subjects factor, confirms no significant effect of Time ($F(1, 18) = .612, p = .442$) or Condition ($F(1, 18) = .898, p = .356$), and no evidence of a Time*Condition interaction ($F(1, 18) = 1.585, p = .224$).’*

We are very grateful for your extremely helpful comments.

===PREPARING YOUR MANUSCRIPT===

At Step 6 'Details & comments', you should review and respond to the queries on the electronic

submission form. In particular, we would ask that you do the following:

-- Ensure that your data access statement meets the requirements at <https://royalsociety.org/journals/authors/author-guidelines/#data>. You should ensure that you cite the dataset in your reference list. If you have deposited data etc in the Dryad repository, please only include the 'For publication' link at this stage. You should remove the 'For review' link.
